# Postoperative Hematoma Expansion in Patients Undergoing Decompressive Hemicraniectomy for Spontaneous Intracerebral Hemorrhage

**DOI:** 10.3390/brainsci12101298

**Published:** 2022-09-26

**Authors:** Martin Vychopen, Johannes Wach, Tim Lampmann, Harun Asoglu, Valeri Borger, Motaz Hamed, Hartmut Vatter, Erdem Güresir

**Affiliations:** Department of Neurosurgery, University Hospital Bonn, 53127 Bonn, Germany

**Keywords:** decompressive hemicraniectomy, intracerebral bleeding, anticoagulant/antiplatelet medication

## Abstract

Introduction: The aim of the study was to analyze risk factors for hematoma expansion (HE) in patients undergoing decompressive hemicraniectomy (DC) in patients with elevated intracranial pressure due to spontaneous intracerebral hematoma (ICH). Methods: We retrospectively evaluated 72 patients with spontaneous ICH who underwent DC at our institution. We compared the pre- and postoperative volumes of ICH and divided the patients into two groups: first, patients with postoperative HE > 6 cm^3^ (group 1), and second, patients without HE (group 2). Additionally, we screened the medical history for anticoagulant and antiplatelet medication (AC/AP), bleeding-related comorbidities, age, admission Glasgow coma scale and laboratory parameters. Results: The rate of AC/AP medication was higher in group 1 versus group 2 (15/16 vs. 5/38, *p* < 0.00001), and patients were significantly older in group 1 versus group 2 (65.1 ± 16.2 years vs. 54.4 ± 14.3 years, *p* = 0.02). Furthermore, preoperative laboratory tests showed lower rates of hematocrit (34.1 ± 5.4% vs. 38.1 ± 5.1%, *p* = 0.01) and hemoglobin (11.5 ± 1.6 g/dL vs. 13.13 ± 1.8 g/dL, *p* = 0.0028) in group 1 versus group 2. In multivariate analysis, the history of AC/AP medication was the only independent predictor of HE (*p* < 0.0001, OR 0.015, CI 95% 0.001–0.153). Conclusion: We presented a comprehensive evaluation of risk factors for hematoma epansion by patients undergoing DC due to ICH.

## 1. Introduction

Spontaneous intracerebral hemorrhage (ICH) is a devastating neurological condition with high mortality and disability rates [1]. The massive effect of the hematoma and swollen tissue can result in elevation of intracranial pressure and subsequently in deterioration and death [2]. Although the standards for best medical treatment are well-established [3], the optimal surgical treatment of the ICH is not clearly defined yet [4] and is a primary endpoint of a running prospective trial (SWITCH). According to contemporary literature, decompressive hemicraniectomy (DC) might be the only option by which to reduce the mortality of patients who fail to respond to conservative treatment [5,6].

Despite of lack of prospective studies, DC is widely used in ultima ratio therapy for refractory high intracranial pressure caused by ICH. One of the most severe complications of the procedure is postoperative expansion of the ICH due to the loss of the tamponade effect [7], resulting in neurological deterioration and poor outcomes, despite the DC [8,9]. The tamponade effect is also one of the strongest arguments not to perform/postpone the surgical therapy [7]. As a result of this, timing of the DC for ICH is still solely individualized compared to other pathologies [10] and can differ significantly among the centers [11]. Among the experimental studies on DC, there is no clear consensus about optimal timing of the procedure [12,13]. In a clinical study of DC on space occupying infarction [14], an early DC was recommended due to an improved functional outcome. Currently, no standard surgical algorithm has been proposed for patients with ICH.

Due to the lack of standards in surgical therapy, the analysis of the hematoma expansion after DC has not been investigated yet. In this study, we present the analysis of the risk factors and short and long-term outcomes in patients with ICH who underwent DC with a focus on postoperative hematoma expansion.

## 2. Materials and Methods

### 2.1. Participants and Study Design

We performed a retrospective analysis of the patients who underwent DC because of spontaneous, non-traumatic ICH at our institution. The patients obtained an external ventricular drain to monitor for ICP. In the case of persistent and therapy refractory intracranial hypertension (ICP > 20 mmHg), DC was performed. All DCs were performed in a standardized fashion: The fully shaved head of the patient was positioned in a Mayfield skull clamp, a question-mark incision was made and a myocutaneous trauma flap was created. After the bone-flap removal, the dura mater was opened in stellar fashion. According to the institutional algorithm and our already published data, we did not perform surgical hematoma evacuation [15]. For closure, we used a rapid closure technique [16].

### 2.2. Exclusion Criteria

Excluded were all patients with insufficient postoperative imaging, traumatic and tumorous ICH and ICH due to vascular malformation.

### 2.3. Neuroimaging

The first postoperative CT-scan (Philips IQon Spectral CT) was performed within 24 h. We used Brain-Lab software—Smartbrush (Brainlab, Feldkirchen, Bavaria, Germany)—to measure the preoperative and postoperative volume of the ICH and pre- and postoperative midline shift (MLS). MLS was measured in millimeters, as the perpendicular distance between the septum pellucidum and the ideal midline. The ideal midline was defined as the line being coplanar with the falx cerebri [17]. The cut for ICH expansion was set to 6 cm^3^ [18]. We dichotomized the study cohort into two groups according to the difference between preoperative and postoperative ICH. Group 1: patients with postoperative expansion of the primary ICH of more than 6 cm^3^. Group 2: patients whose ICH size difference was none or smaller than 6 cm^3^.

The equation proposed by DeBonis [19] was used to calculate the surface of the craniectomy. The proposed formula used a variation on the spherical cap formula Ab=(π/4)dD, where Ab represents the surface of the craniectomy, d anteroposterior and D craniocaudal diameder of the craniectomy. The measurement was performed in IMPAX FX AFGA Healthcare. Each patient’s medical history was screened for anticoagulant and antiplatelet medication (AC/AP medication), comorbidities such as arterial hypertension and chronic kidney failure, age, and Glasgow coma scale (GCS) at the time of admission. Platelet concentrate and tranexamic acid were routinely administered by patients with a history of anticoagulants/antiplatelet medication (AC/AP medication). The admission laboratory tests were examined to collect data for blood particles counts (red blood cells, platelets), levels of antithrombin and other coagulation factors (I, II, V, VII, VIII, X), platelet function assay test (aPTT) and Quick test. In cases of pathological results, an additional factor substitutions scheme was discussed with the attending hematologist. Due to the emergency nature of the procedure, we did not perform a second preoperative control for the laboratory tests after substitution was administered. Although not following a precisely defined scheme, we generally performed very aggressive substitution prior to skin incision.

### 2.4. Hypertension Management

Patients who showed acute hypertensive crisis on admission were treated with Urapidil i.v. to reach the systolic blood pressure niveau < 140 mmHg. In the course of the hospitalization, a long-term blood pressure management and medication plan was optimized by the attending cardiologist.

### 2.5. Statistics

We used IMB SPSS Statistics (Version 27) to perform statistical analysis. A Chi-square test was used for univatiate analysis of the risk-factors. One sample T test for a mean with standard deviation was used for statistical evaluation of continuous variables. For groups with n < 10, Yates correction for continuity was used to avoid an approximation error. All results with *p* < 0.05 were considered as statistically significant.

Variables that reached statistical significance were subsequently used to perform multivariate regression analysis.

## 3. Results

We retrospectively analyzed 72 patients with ICH who underwent DC at our institution. Eighteen patients were excluded because of the following: intraoperative blood-clot evacuation (n = 12), insufficient postoperative imaging (n = 2), tumorous ICH (n = 2) and ICH from arteriovenous malformation (n = 2), leaving 54 patients for the analysis (Table 1).

### 3.1. Preoperative/Postoperative Size Differences

The mean size difference in the Group 1 was significantly higher, at 65.1 ± 47.9 cm^3^. Group 2 showed a size difference of −0.5 ± 4.2 cm^3^ (*p* < 0.0001). For an example of a patient with ICH expansion, see Figure 1. For an example of a patient without ICH expansion, see Figure 2.

### 3.2. Anticoagulant/Antiplatelet Medication (AC/AP)

Fifteen out of 16 patients in Group 1 had a positive medical history for AC/AP medication (93.7%). There was significantly lower AC/AP usage in Group 2 (5/38, 13.1%, *p* < 0.00001).

### 3.3. Age

The patients with postoperative ICH expansion were on the average 12 years older (65.1 ± 16.2 years vs. 54.4 ± 14.3 years, *p* = 0.0193).

### 3.4. Blood Particles

Group 1 showed lower rates of both hematocrit (34.1 ± 5.4% vs. 38.1 ± 5.1%, *p* = 0.0125) and hemoglobin (11.5 ± 1.6 g/dL vs. 13.13 ± 1.8 g/dL, *p* = 0.0028). There was no difference in platelet count (219.2 ± 83 G/L vs. 194.0 ± 88 G/L, *p* = 0.3333).

### 3.5. Coagulation Parameters

We did not observe any differences in preoperative quick analysis; aPTT test; antithrombin analysis; analysis of factors I, II, V, VII, VIII and X; or platelet function assay test. For detailed information, see Table 1.

### 3.6. Comorbidities

Twelve out of 16 patients in Group 1 had diagnosed arterial hypertension compared to 30/38 patients in Group 2. Two/three patients showed elevated creatinine levels. We did not note any genetic clotting disorders among the patient collective.

### 3.7. Mid-line Shift and DC Size

There was a homogeneous distribution of both preoperative (11.7 ± 5.8 mm vs. 10.3 ± 5.5 mm, *p* = 0.4) and postoperative (6.9 ± 4.8 mm vs. 5.8 ± 5.4 mm, *p* = 0.4) mid-line shift among the groups. The size of the DC did not correlate with postoperative hematoma expansion (218 ± 50.2 cm^3^ vs. 204.4 ± 44.9 cm^3^, *p* = 0.35).

### 3.8. Outcome According to Modified Rankin Scale (mRS) 6 Months Postoperatively

Patients with ICH expansion showed a worse 30-day hospital mortality and 6-month outcome according to the mRS. We observed no patient with a positive outcome in Group 1, as opposed to 10 patients with a positive outcome according to mRS in Group 2 (*p* < 0.0001). Additionally, 30-day hospital mortality in Group 1 was 87.5% vs. 28.9% in Group 2 (*p* < 0.0001).

### 3.9. Multivariate Analysis of Univariate Significant Risk Factors

Multivariate progressive regression analysis was performed, considering the level of hemoglobin and hematocrite at admission, age and medical history of AC/AP. The only significant and independent predictor of ICH expansion was the AC/AP medication.

### 3.10. AC/AP Medication as an Outcome and Mortality Predictor

Compared to the control group, patients with AC/AP medication showed higher 30-day mortality (17/20, 85% vs. 9/34, 26.4%. *p* = 0.0001) regardless of hematoma expansion. We found no patient with a long-term positive outcome and AC/AP medication in the medical history compared to 10/34 patients with mRS ≤ 3.

## 4. Discussion

We performed a retrospective evaluation of risk factors for postoperative expansion of ICH by patients who underwent DC as a therapy for intracranial hypertension. Our univariate analysis showed that ICH expansion after DC has similar risk factors to those already published for primary ICH [20]: AC/AP medication, hypertension, lower hemoglobin and hematocrite level and age. However, the multivariate analysis showed that AC/AP medication is the only significant expansion predictor.

In the general population, the number of patients treated with long-term oral AC/AP medication is increasing [21,22]. Being the most devastating complication of this therapy, ICH therapeutic algorithms, including hemostatic therapy and blood-pressure management, are being developed. For the hemostatic therapy, many standardized substitution schemes have already been proposed, including tranexamic acid, desmopressin and platelet transfusion [23] and recombinant activated factor substitution [24]. Moreover, the long-term benefit of hemostatic therapy seems to be controversial [20]. Due to the inconclusive data for hemostatic therapy, we used a strongly individualized approach to substitution-therapy based on the multidisciplinary agreement. Although the hemostatic therapy in our cohort was always performed under hematological supervision and in accordance with the guidelines for reversal antithrombotic in intracranial hemorrhage [25], there was no standardized study protocol or scheme. The definite art and dosage of hemostatic therapy were solely dependent on the consensus between attending hematologist, neurologist and neurosurgeon. Despite the aggressive substitution, AC/AP medication remained the strongest risk factor for hematoma expansion. A clear algorithm for preoperative substitution might prevent this fatal complication. However, for the development of such an algorithm, a multicentric design is necessary due to the rarity of this specific condition.

The size of initial ICH was already described as a risk factor for hematoma expansion by patients treated conservatively [26,27]. However, DC was only performed in the patient with a large ICH volume as ultima ratio therapy for the treatment of elevated intracranial pressure. Due to this strong bias, we did not include the preoperative size of the ICH in our multivariate analysis. Compared to Fung et al. [6], we found similar means of preoperative hematoma size (61.3 cm^3^ vs. 67.8 cm^3^).

The expansion/blossoming of an already present hemorrhage after DC has been associated with poor postoperative outcomes. A loss of tamponade effect was proposed by Gopalakrishnan et al. [7] as a reason for hematoma expansion after DC. After the removal of the bone flap, the reduced ICP facilitates the ipsilateral hematoma expansion and/or blossoming of the hemorrhagic lesion [7,28,29]. This effect could be potentially compounded by the size of the DC. Qui et al. and Jiang et al. both examined the effect of the size of the decompression and suggested higher incidence of postoperative ipsilateral hematomas by larger craniectomies [30,31]. In more contemporary literature, the same DC size to hemorrhage correlation was proposed by Cepeda et al. [32] Surprisingly, we found no relation in the size of the craniectomy and the hematoma expansion, which indirectly contradicts the tamponade effect. This would have a strong impact on procedure timing, which according to contemporary literature, varies among the different cohorts between 4 and 96 h from symptom onset [11]. Since our data indirectly contradict the tamponade effect, the ICH patient might also benefit from early craniectomy. This statement is strongly limited by the retrospective design of the study and missing experimental data to support/contradict the tamponade effect in ICH patients.

Although a higher blood pressure baseline was described to predict HE [33], no correlation between hypertension and HE was observed in our group. This might have been due to our aggressive blood pressure management with a strict systolic blood pressure target of <140 mmHg.

Our data showed the devastating effect of the hematoma expansion. This finding might support the possible consideration of changing the therapy goal to best support care due to the poor prognosis and very high 30-day mortality. However, due to the retrospective design of our study, all such decisions had to be made solely on individual basis.

Additionally, our data suggest that AC/AP medication itself seems to be a very strong mortality and outcome predictor for patients undergoing DC.

## 5. Limitations

Our study had several limitations. The main limitation was the retrospective design. Furthermore, we used no standardized protocol facilitating the decision making process regarding DC. Due to the rarity of the condition, the sample size was a strong limitation of our study. Although we generally performed aggressive correction of any present coagulopathy, there was no clearly defined protocol for the substitution scheme. Although we performed coagulation-factors substitution under supervision of hematology colleagues, the individual factor-substitution scheme depended on the attending hematologist.

## 6. Conclusions

In this study, we presented a comprehensive evaluation of postoperative hematoma expansion in patients undergoing DC. Hematoma expansion after DC for intracerebral hemorrhage is associated with antiplatelet/anticoagulant medications.

## Figures and Tables

**Figure 1 brainsci-12-01298-f001:**
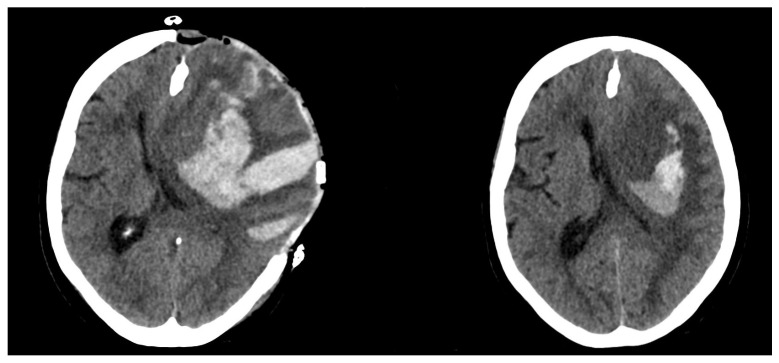
Postoperative (**left**)/Preoperative (**right**) CT scan of a patient with hematoma expansion.

**Figure 2 brainsci-12-01298-f002:**
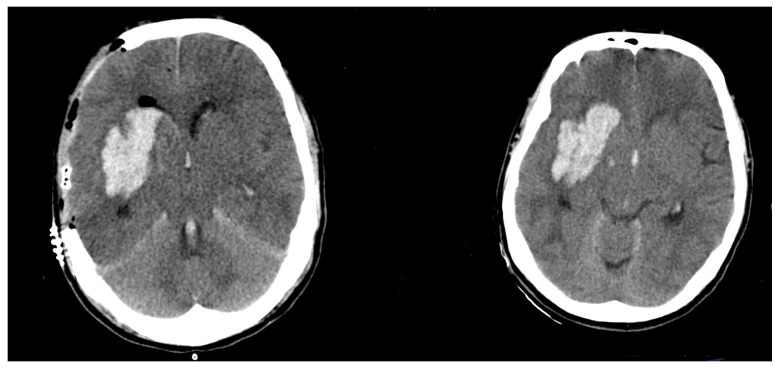
Postoperative (**left**)/Preoperative (**right**) CT scan of a patient without hematoma expansion).

**Table 1 brainsci-12-01298-t001:** Patient characteristics.

	ICH Expansion < 6 cm^3^ (No Expansion) n = 16	ICH Expansion >= 6 cm^3^ (Expansion) n = 38	*p* Value
ICH size preoperativ	87.1 ± 40.8 cm^3^	59.72 ± 27.4 cm^3^	n.s.
size difference	65.1 ± 47.9 cm^3^	−0.5 ± 4.19 cm^3^	*p* < 0.0001
Midline shift preop.	11.7 ± 5.8 mm	10.3 ± 5.5 mm	n.s.
Midline shift postop.	6.9 ± 4.8 mm	5.8 ± 5.4 mm	n.s.
AC/AP Medication (%)	15/16 (93.75%)	5/38 (13.15%)	*p* < 0.00001
DC Size	218.7 ± 50.2 cm^2^	204.4 ± 44.92 cm^2^	n.s.
Intraoperative blood loss	980 ± 804 mL	784 ± 520 mL	n.s.
mRS positive	0/16 (0%)	10/38 (26.3%)	*p* < 0.0001
30-day mortality	15/16 (93.75%)	11/38 (28.9%)	*p* < 0.0001
Age (years)	66.7 ± 16.2	54.4 ± 12.4	*p* = 0.0193
**Laboratory results**			
Platelets	219.2 ± 83	194.0 ± 88	n.s.
Hematocrit (%)	34.1 ± 5.4	38.1 ± 4.8	*p* = 0.0095
Hemoglobin (g/dL)	11.5 ± 1.6	13.13 ± 1.8	*p* = 0.0028
Pathological creatinine /Chronic kid-ney failure	2/16	3/38	n.s.
Tranexamic Acid + FFP + Prothrom-bin substitution	14/16	12/38	n.s.
Pathological Quick	6/16	10/38	n.s.
**Single factor analysis**			
Antitrombin <85%	7/16	19/38	n.s.
F I < (ref. 180–355 mg/dL)	1/16	6/38	n.s.
F II < (ref. 83–145%)	7/16	6/38	n.s.
F V < (ref. 75–152%)	4/16	7/38	n.s.
F VII < (ref. 74–158%)	5/16	10/38	n.s.
F VIII < (ref. 67–220%)	0/38	0/38	n.s.
F X < (ref. 80–140%)	2/16	14/38	n.s.
PFA Test positive	8/16	15/38	n.s.
Hypertension	12/16	30/38	n.s. ^1^

^1^ ICH—intracerebral hemorrhage, AC/AP—anticoagulant/antiplatelet, DC—decompressive craniectomy, mRS—modified Rankin scale, F I—factor I, F II—factor II, F V—factor V, F VII—factor VII, F VIII—factor VIII, F X—factor X, PFA—platelet function assay test, n.s.—not significant.

## Data Availability

Not applicable.

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
