# Peer review of "Postoperative Hematoma Expansion in Patients Undergoing Decompressive Hemicraniectomy for Spontaneous Intracerebral Hemorrhage"

_brainsci, 2022, doi:10.3390/brainsci12101298_

Round 1

Reviewer 1 Report

The authors studied Postoperative hematoma expansion in patients undergoing decompressive hemicraniectomy for spontaneous intracerebral hemorrhage

it is an interesting study:

avoid using abbreviations in the keyword

The introduction is relatively short and is to be extended.

Avoid mentioning and describing results in Methods  ( figure 1 and 2 )

The methodology needs to be described with more detail: The equation proposed by DeBonis ?/ CT scan machine ?/ application used for the measurements? /  timing of the postoperative CT scan? more details on the surgical technique: opening of the dura mater?

abbreviations must be mentioned from their first appearance in the manuscript ( i.v. ; aPTT test; mRS)

Avoid mentioning and describing Methods in results (We dichotomized the study cohort into two groups according to the difference between preoperative and postoperative ICH; Group 1. Patients with postoperative expansion of the primary ICH of more than 6 cm3 and Group 2. Patients whose ICH size difference was none or smaller than 6 cm3)

Numbers at the beginning of the sentence must be written in letters  (12 out of……/ resp. 3 patients showed …..

there is a meaningless phrase: Insert Table 2

there is no table 2 ?!

Have you checked the repetitiveness of the results?

The numerical part is not well prepared and presented/ Table 1 is very difficult to read

The discussion is relatively short and is to be extended.

Avoid mentioning and describing Results in the discussion

 Some paragraphs in the discussion section are without references

The English level is to be improved.

Author Response

Author’s Revision Letter

Dear Reviewer,

we would like to thank you for thoroughly reviewing our manuscript and your comments! We have addressed each of the concerns mentioned point-by-point below and hope that they are adequate in this regard.

The authors studied Postoperative hematoma expansion in patients undergoing decompressive hemicraniectomy for spontaneous intracerebral hemorrhage

it is an interesting study:

1. Avoid using abbreviations in the keyword

We revised the key-word as follows:

decompressive hemicraniectomy, intracerebral bleeding, Anticoagulant/Antiplatelet medication”

2.The introduction is relatively short and is to be extended.

Thank you for the tip. We have extendded the introduction part to stress out the running SWITCH trial and tamponade effect controversy:

Spontaneous intracerebral hemorrhage (ICH) is a devastating neurological condition with a high mortality and disability rates \cite{1}. The mass effect of the hematoma and swollen tissue can result in elevation of intracranial pressure and subsequently in deterioration and death \cite{2}. For patients who fail to respond to conservative treatment, early decompressive hemicraniectomy (DC) might be the only option to reduce the mortality \cite{3,4}. Although the standards for best medical treatment are well-established \cite{5}, the optimal surgical treatment of the ICH is not clearly defined yet \cite{6} and is a primary endpoint of a running prospective trial (SWITCH). Despite of lack of prospective studies, DC is widely used as ultima ratio therapy by refractory high intracranial pressure. However, in case of surcigal therapy, the timing of the procedure is still solely individualized and can differ significantly among the different centers \cite{24}
One of the possible complications of the procedure is postoperative expansion of the ICH due to the loss of the tamponade effect \cite{7} resulting in neurological deterioration and poor outcome despite the DC \cite{8,9}.
The tamponade effect is also one of the strongest arguments not to perform/postpone the surgical therapy\cite{7}.
Among the experimental studies on DC, there is no clear consensus about optimal timing of the procedure \cite{25,26}. In presented clinical study for DC on space occupying infarction \cite{27}, an early DC is recommended due to improved functional outcome. Up to date, no standard surgical algorithm has been proposed for patients with ICH.
Because of the lack of standards in surgical therapy, the analysis of the hematoma expansion after DC has not been investigated yet. In the present study, we focus on analysis of the risk factors for postoperative hematoma expansion in patients with ICH who underwent decompressive craniectomy.

Added references:

\bibitem{24} Morgenstern LB., American Heart Association Stroke Council and Council on Cardiovascular Nursing. Guidelines for the management of spontaneous intracerebral hemorrhage: a guideline for healthcare professionals from the American Heart Association/American Stroke Association. {\em Stroke.} {\bf 2010}, {\em Sep}, 41(9):2108-29.

\bibitem{25} Forsting M., Decompressive craniectomy for cerebral infarction. An experimental study in rats. {\em Stroke.} {\bf 1995}, {\em Feb}, 26(2):259-64.

\bibitem{26} Doerfler A., Decompressive craniectomy in a rat model of "malignant" cerebral hemispheric stroke: experimental support for an aggressive therapeutic approach. {\em J Neurosurg.} {\bf 1996}, {\em Nov}, 85(5):853-9.

\bibitem{27} Hofmeijer J., Surgical decompression for space-occupying cerebral infarction (the Hemicraniectomy After Middle Cerebral Artery infarction with Life-threatening Edema Trial [HAMLET]): a multicentre, open, randomised trial. {\em Lancet Neurol.} {\bf 2009}, {\em Apr}, 8(4):326-33.

3. Avoid mentioning and describing results in Methods  ( figure 1 and 2 )

We moved the figure 1 and 2 to our result part to demonstrate the results in Result-part of the manuscript.

4. The methodology needs to be described with more detail: 

The equation proposed by DeBonis ?/ CT scan machine ?/ application used for the measurements? /  timing of the postoperative CT scan? more details on the surgical technique: opening of the dura mater?

4.1 To more clearly state our Methodology, we added following explanation of the DC measurement:

The equation proposed by DeBonis \cite{14} was used to calculate the surface of the craniectomy. The proposed formula used a variation on spherical cap formula \begin{math}Ab = (\pi/4)d D\end{math}, where Ab represents the surface of the craniectomy, d anteroposterior and D craniocaudal diameder of the craniectomy. The measurement was performed in IMPAX FX AFGA Healthcare.

4.2. For sufficient description of our surgical algorithm, we added following paragraph in our manuscript:

We performed a retrospective analysis of the patients who underwent decompressive hemicraniectomy (DHC) because of spontaneous, non-traumatic ICH at our institution. All DHCs were performed in a standardized fashion. The fully shaved head of the patient is positioned in a mayfield-skull clamp, a questionmark inscision is performed and myocutaneous trauma flap is created. After the bone-flap removal, the dura-mater is opend in stellate fashion. Subsequently, rapid-closure technique \cite{10} is performed without surgical hematoma \cite{10,11}. The first postoperative CT-Scan is performed within 24 hours.

4.3. CT-scan machine:

The first postoperative CT-scan (Philips IQon Spectral CT) is performed within 24 hours.

5. abbreviations must be mentioned from their first appearance in the manuscript ( i.v. aPTT testmRS)

Thank you for the recommendation, we corrected the abbreviations according to your proposal.

6. Avoid mentioning and describing Methods in results (We dichotomized the study cohort into two groups according to the difference between preoperative and postoperative ICH; Group 1. Patients with postoperative expansion of the primary ICH of more than 6 cm3 and Group 2. Patients whose ICH size difference was none or smaller than 6 cm3)

We moved the paragraph from Results into Method section.

7. Numbers at the beginning of the sentence must be written in letters  (12 out of……resp. 3 patients showed …..

We corrected this spelling mistake in following sentences:

Fifteen out of 16 patients in Group 1 had a positive medical history for AC/AP medication (93.7\%).

Twelve out of 16 patients in 1 group had diagnosed arterial hypertension compared to 30/38 patients in N group.

Two, resp. 3 patients showed elevated creatinine level.

8. There is a meaningless phraseInsert Table 2

there is no table 2 ?!

We deleted the meaningless phrase. This mistake happened due to changes in manuscript format before submitting the final version.

Have you checked the repetitiveness of the results?

We tried to avoid the repetitiveness, mainly through reducing the count of phrase “statistically significant”. Thank you for this comment.

The numerical part is not well prepared and presented/ Table 1 is very difficult to read

Thank you for the comment. We see, that the intense row of numbers is not the optimal way to present our results. To make the Table 1 easier for the reader, we changed the p values > 0.05 to n.s. (not significant). Moreover, we added two Table subsections (Laboratory results and single factor analysis) to make it easier for the reader to follow.

The discussion is relatively short and is to be extended. Avoid mentioning and describing Results in the discussion. Some paragraphs in the discussion section are without references.

To extend the discussion, we added following parts:

AC/AP Medication

In general population, the number of patients treated with long-term oral AC/AP medication is increasing \cite{22,23}. Being the most devastating complication of this therapy, ICH therapeutic algorithms including hemostatic therapy and blood-pressure management are being developed.

Tamponade effect:

Contrary to the proposed tamponade effect which is usually among the strongest arguments for postponing the surgical procedure \cite{7}, we found no relation in the size of the craniectomy and the hematoma expansion, which indirectly contradicts the tamponade effect. This would have strong impact on procedure timing, which according to contemporary literature varies among the different cohorts between 4 to 96 hours from symptom onset \cite{24}. Since our data indirectly contradict the tamponade effect, the ICH patient might also benefit from early craniectomy. This statement is strongly limited by the retrospective design of the study and missing experimental data to support/contradict the tamponade effect in ICH patients.

Outcome

Our data showed devastating effect of the hematoma expansion. This finding might support the possible consideration of changing the therapy goal to best supportive care due to the poor prognosis and very high 30-day mortality. However, due to the retrospective design of our study, all such decisions have to be done solely on individual basis.

The references were corrected among the manuscript. Now, all references should be displayed correctly. Furthermore, following references were added:

\bibitem{22} Alcusky M., Changes in Anticoagulant Utilization Among United States Nursing Home Residents With Atrial Fibrillation From 2011 to 2016. {\em J Am Heart Assoc.} {\bf 2019}, {\em May}, 8(9):e012023.

\bibitem{23} Campitelli MA., Trends in Anticoagulant Use at Nursing Home Admission and Variation by Frailty and Chronic Kidney Disease Among Older Adults with Atrial Fibrillation. {\em Drugs Aging.} {\bf 2021}, {\em Jul}, 38(7):611-623.

\bibitem{24} Morgenstern LB., American Heart Association Stroke Council and Council on Cardiovascular Nursing. Guidelines for the management of spontaneous intracerebral hemorrhage: a guideline for healthcare professionals from the American Heart Association/American Stroke Association. {\em Stroke.} {\bf 2010}, {\em Sep}, 41(9):2108-29.

The English level is to be improved.

We kindly asked MDPI to perform an English check.

Again, we would like to thank all reviewers for thoroughly reading the manuscript and for the critics, the hints to the weak points and suggestions to improve our manuscript.

Yours sincerely,

Martin Vychopen, M.D.

(on behalf of the authors)

Reviewer 2 Report

This is a very interesting paper focused on predictors of postoperative hematoma expansion in patients undergoing decompressive hemicraniectomy for spontaneous intracerebral hematoma. The paper is well-written and of interest for the readers, however, several minor changes should be made before considering it for publication.

Abstract.

1- The sample was divided into two groups according to the volume of the hematoma expansion. Why was 6 cm3 used as a cutt off to divide the sample?

2- The conclusion of the abstract is once again mentioning the result of the statistically significant factors contributing to the hematoma expansion in multivariate analyses. Please, do not repeat the results and build a conclusion based on the main results.

Introduction.

1- The introduction is brief. I recommend to expand it by explaining, firstly, which are the main factors contributing to the onset of spontaneous intracerebral hemorrhage, which is the focus of the study. Please, add some more new references.

2- The main aim of this study should be better explained at the end of the introduction section. What are the main hypotheses the authors are exploring.

Methods

1- This section should be divided into several subsections. 1. Participants and study design. 2. Inclusion and exclusion criteria. 3. Neuroimaging- a further description of the methods used and the technique are expected.

2- More details about statistical analyses should be described.

Results.

1- The sample is divided into two groups which are seemingly unbalanced. Could it be a potential bias or confounder for the main findings? Please, add it in the limitations section.

2- Are non-parametric tests needed for comparison purposes?

Discussion

1- The discussion section is really brief. I would recommend to expand it by discussing factors potentially and not potentially explaining hematoma expansion. 

Conclusions

1- There is no specific conclusions section. I recommend to add a subsection or section after the limitations section. Please di not simply mention the main finding of the study.

Limitations

1- The unbalanced sample size should be also added as a limitation. 

2- What about mentioning the main strengths of the study?

Author Response

Author’s Revision Letter

Dear Reviewer,

we would like to thank you for thoroughly reviewing our manuscript and your comments! We have addressed each of the concerns mentioned point-by-point below and hope that they are adequate in this regard.

This is a very interesting paper focused on predictors of postoperative hematoma expansion in patients undergoing decompressive hemicraniectomy for spontaneous intracerebral hematoma. The paper is well-written and of interest for the readers, however, several minor changes should be made before considering it for publication.

Abstract.

1- The sample was divided into two groups according to the volume of the hematoma expansion. Why was 6 cm3 used as a cutt-off to divide the sample?

The cut-off for the sample was indeed very intensely discussed among the authors. After carefull consideration, we did choose 6cm3 according to definition of hematoma expansion by Bolouis et al (2016, JAMA-Neurology). However, first draft of the manuscript used the 10 cm3 cut-of without change of significant results and the results of multivariate analysis.

2- The conclusion of the abstract is once again mentioning the result of the statistically significant factors contributing to the hematoma expansion in multivariate analyses. Please, do not repeat the results and build a conclusion based on the main results.

We tried to avoid repeating the results and re-formulated our Abstract-conclusion. Further revision of Conclusion part of the manuscript are mentioned bellow:

Conclusion: We presented a comprehensive evaluation of risk factors for hematoma epansion by patients undergoing DC due to ICH.

Introduction.

1- The introduction is brief. I recommend to expand it by explaining, firstly, which are the main factors contributing to the onset of spontaneous intracerebral hemorrhage, which is the focus of the study. Please, add some more new references.

Thank you for this comment. After carefull consideration, we expanded the introduction in 3 main points: 1. SWITCH study, an ongoing prospective trial on DC in ICH patients was mentioned, 2. the controversy of timing is mentioned and 3. the tamponade effect is explained and subsequently expanded in the discussion part:

Spontaneous intracerebral hemorrhage (ICH) is a devastating neurological condition with a high mortality and disability rates \cite{1}. The mass effect of the hematoma and swollen tissue can result in elevation of intracranial pressure and subsequently in deterioration and death \cite{2}. For patients who fail to respond to conservative treatment, early decompressive hemicraniectomy (DC) might be the only option to reduce the mortality \cite{3,4}. Although the standards for best medical treatment are well-established \cite{5}, the optimal surgical treatment of the ICH is not clearly defined yet \cite{6} and is a primary endpoint of a running prospective trial (SWITCH). Despite of lack of prospective studies, DC is widely used as ultima ratio therapy by refractory high intracranial pressure. However, in case of surcigal therapy, the timing of the procedure is still solely individualized and can differ significantly among the different centers \cite{24}
One of the possible complications of the procedure is postoperative expansion of the ICH due to the loss of the tamponade effect \cite{7} resulting in neurological deterioration and poor outcome despite the DC \cite{8,9}.
The tamponade effect is also one of the strongest arguments not to perform/postpone the surgical therapy\cite{7}.
Among the experimental studies on DC, there is no clear consensus about optimal timing of the procedure \cite{25,26}. In presented clinical study for DC on space occupying infarction \cite{27}, an early DC is recommended due to improved functional outcome. Up to date, no standard surgical algorithm has been proposed for patients with ICH.
Because of the lack of standards in surgical therapy, the analysis of the hematoma expansion after DC has not been investigated yet. In the present study, we focus on analysis of the risk factors for postoperative hematoma expansion in patients with ICH who underwent decompressive craniectomy.

Added references:

\bibitem{24} Morgenstern LB., American Heart Association Stroke Council and Council on Cardiovascular Nursing. Guidelines for the management of spontaneous intracerebral hemorrhage: a guideline for healthcare professionals from the American Heart Association/American Stroke Association. {\em Stroke.} {\bf 2010}, {\em Sep}, 41(9):2108-29.

\bibitem{25} Forsting M., Decompressive craniectomy for cerebral infarction. An experimental study in rats. {\em Stroke.} {\bf 1995}, {\em Feb}, 26(2):259-64.

\bibitem{26} Doerfler A., Decompressive craniectomy in a rat model of "malignant" cerebral hemispheric stroke: experimental support for an aggressive therapeutic approach. {\em J Neurosurg.} {\bf 1996}, {\em Nov}, 85(5):853-9.

\bibitem{27} Hofmeijer J., Surgical decompression for space-occupying cerebral infarction (the Hemicraniectomy After Middle Cerebral Artery infarction with Life-threatening Edema Trial [HAMLET]): a multicentre, open, randomised trial. {\em Lancet Neurol.} {\bf 2009}, {\em Apr}, 8(4):326-33.

2- The main aim of this study should be better explained at the end of the introduction section. What are the main hypotheses the authors are exploring.

We hope to have adressed the point in previous response. Up-to date, there is no similar study published on patients with DC. The main reason for this is lack of standard therapy. The main aim of our study was to descriptively analyze the risk factors and indirectly examine the tamponade effect.

Methods

1- This section should be divided into several subsections. 1. Participants and study design. 2. Inclusion and exclusion criteria. 3. Neuroimaging- a further description of the methods used and the technique are expected.

We divided the Methods part into sections and added detailes on: 1. surgical technique, 2. exclusion criteria, 3. dichotomisation and cut-of, 4. the surface-calculation formula.\subsection{Participants and study design}
We performed a retrospective analysis of the patients who underwent DC because of spontaneous, non-traumatic ICH at our institution. All DCs were performed in a standardized fashion:
The fully shaved head of the patient is positioned in a mayfield-skull clamp, a questionmark inscision is performed and myocutaneous trauma flap is created. After the bone-flap removal, the dura-mater is opend in stellate fashion. For closure, we use rapid-closure technique \cite{10} without surgical hematoma evacuation \cite{11}.
\subsection{Exclusion criteria}
Excluded were all patients with insufficient postoperative imaging, traumatic and tumorous ICH and ICH due to vascular malformation.

\subsection{Neuroimaging}
The first postoperative CT-scan is performed within 24 hours. We used Brain-Lab software - Smartbrush (Brainlab, Feldkirchen, Bavaria, Germany) to measure preoperative and postoperative volume of the ICH and pre- and postoperative midline shift (MLS). MLS was measured in millimeters, as the perpendicular distance between the septum pellucidum and the ideal midline. Ideal midline was defined as the line being coplanar with the falx cerebri \cite{12}. The cut of for ICH expansion was set on 6 cm3 \cite{13}.
We dichotomized the study cohort in two groups according to the difference between preoperative and postoperative ICH; Group 1. Patients with postoperative expansion of the primary ICH of more than 6 cm3 and Group 2. Patients whose ICH size difference was none or smaller than 6 cm3.
The equation proposed by DeBonis \cite{14} was used to calculate the surface of the craniectomy.
The proposed formula used a variation on spherical cap formula \begin{math}Ab = (\pi/4)d D\end{math}, where Ab represents the surface of the craniectomy, d anteroposterior and D craniocaudal diameder of the craniectomy. The measurement was performed in IMPAX FX AFGA Healthcare. Patient’s medical history was screened for anticoagulant and antiplatelet medication (AC/AP medication), comorbidities such as arterial hypertension and chronic kidney failure, age, and Glasgow coma scale (GCS) at the time of admission. Platelet concentrate and tranexamic acid were routinely administered by patients with history of anticoagulants/antiplatelet medication (AC/AP medication). The admission laboratory tests were examined to collect data for blood particles counts (red blood cells, platelets), levels of antithrombin and other coagulation factors (I, II, V, VII, VIII, X), platelet function assay test (aPTT) and Quick test. In case of pathological results, additional factor substitutions scheme was discussed with the attending hematologist.

2- More details about statistical analyses should be described.

We reorganised the statistical part as follows:

\subsection{Statistics}
We used IMB SPSS Statistics (Version 27) to perform T-Test, chi-square test for univatiate analysis of the risk-factors. For groups with n<10, Yates correction for continuity was used to avoid an approximation error. All results with p < 0.05 were considered as statistically significant.
Variables that reached statistical significance were subsequently used to perform multivariate regression analysis.

Results.

1- The sample is divided into two groups which are seemingly unbalanced. Could it be a potential bias or confounder for the main findings? Please, add it in the limitations section.

Thank you for this comment. From our point of view, the disbalance between the groups is due to the rarity of this condition (ICH treated conservatively is more easy to analyze compared to ICH undergoing DC). This fact strongly limits our sample size, which indeed is a bias. The other bias of the results is already mentioned in the discussion (we only perform DC in patients with already large volume of ICH).

We added following sentence in our limitation:

Due to the rarity of the condition, the sample size is a strong limitation of our study.

2- Are non-parametric tests needed for comparison purposes?

Due to outcome dichotomisation (mRS), we do not need any non-parametric tests.

Discussion

1- The discussion section is really brief. I would recommend to expand it by discussing factors potentially and not potentially explaining hematoma expansion. 

Thank you for this comment. Indeed, the discussion is very brief and does not include crucial aspects of our findings. We expanded the discussion focusing on the timing of the procedure/tamponade effect as well as the implication of our data about the prognosis.

AC/AP Medication

In general population, the number of patients treated with long-term oral AC/AP medication is increasing \cite{22,23}. Being the most devastating complication of this therapy, ICH therapeutic algorithms including hemostatic therapy and blood-pressure management are being developed.

Tamponade effect:

Contrary to the proposed tamponade effect which is usually among the strongest arguments for postponing the surgical procedure \cite{7}, we found no relation in the size of the craniectomy and the hematoma expansion, which indirectly contradicts the tamponade effect. This would have strong impact on procedure timing, which according to contemporary literature varies among the different cohorts between 4 to 96 hours from symptom onset \cite{24}. Since our data indirectly contradict the tamponade effect, the ICH patient might also benefit from early craniectomy. This statement is strongly limited by the retrospective design of the study and missing experimental data to support/contradict the tamponade effect in ICH patients.

Outcome

Our data showed devastating effect of the hematoma expansion. This finding might support the possible consideration of changing the therapy goal to best supportive care due to the poor prognosis and very high 30-day mortality. However, due to the retrospective design of our study, all such decisions have to be done solely on individual basis.

The references were corrected among the manuscript. Now, all references should be displayed correctly. Furthermore, following references were added:

\bibitem{22} Alcusky M., Changes in Anticoagulant Utilization Among United States Nursing Home Residents With Atrial Fibrillation From 2011 to 2016. {\em J Am Heart Assoc.} {\bf 2019}, {\em May}, 8(9):e012023.

\bibitem{23} Campitelli MA., Trends in Anticoagulant Use at Nursing Home Admission and Variation by Frailty and Chronic Kidney Disease Among Older Adults with Atrial Fibrillation. {\em Drugs Aging.} {\bf 2021}, {\em Jul}, 38(7):611-623.

Conclusions

1- There is no specific conclusions section. I recommend to add a subsection or section after the limitations section. Please do not simply mention the main finding of the study.

In following paragraph, we tried to formulate our conclusion and result application:

Conclusion

In this study, we present a comprehensive evaluation of postoperative hematoma expansion in patients undergoing DC. The outcome analysis might help clinitians to clearly state a realistic therapy goal. This is strongly limited by the design of the study.

Limitations

1- The unbalanced sample size should be also added as a limitation. 

As mentioned above, we added this limitation.

2- What about mentioning the main strengths of the study?

As presented in conclusion. The strength of this study is the evaluation of risk factors by this subgroup of patients and both short and long-term outcome analysis, which might facilitate the therapy-decision.

Again, we would like to thank all reviewers for thoroughly reading the manuscript and for the critics, the hints to the weak points and suggestions to improve our manuscript.

Yours sincerely,

Martin Vychopen, M.D.

(on behalf of the authors)